# The Complete Mitochondrial Genome of *Paeonia lactiflora* Pall. (Saxifragales: Paeoniaceae): Evidence of Gene Transfer from Chloroplast to Mitochondrial Genome

**DOI:** 10.3390/genes15020239

**Published:** 2024-02-14

**Authors:** Pan Tang, Yang Ni, Jingling Li, Qianqi Lu, Chang Liu, Jinlin Guo

**Affiliations:** 1Key Laboratory of Characteristic Chinese Medicine Resources in Southwest China, College of Pharmacy, Chengdu University of Traditional Chinese Medicine, Chengdu 611137, China; tangpan@stu.cdutcm.edu.cn; 2Center for Bioinformatics, Institute of Medicinal Plant Development, Chinese Academy of Medical Sciences and Peking Union Medical College, No. 151, Malianwa North Road, Haidian District, Beijing 100093, China; ny_work@126.com (Y.N.); lijingling1997@163.com (J.L.); lqq1874@126.com (Q.L.); 3College of Medical Technology, Chengdu University of Traditional Chinese Medicine, Chengdu 611137, China

**Keywords:** *P. lactiflora*, de novo assembly, organelle genome, DNA transfer, RNA-editing events

## Abstract

*Paeonia lactiflora* (*P. lactiflora*), a perennial plant renowned for its medicinal roots, provides a unique case for studying the phylogenetic relationships of species based on organelle genomes, as well as the transference of DNA across organelle genomes. In order to investigate this matter, we sequenced and characterized the mitochondrial genome (mitogenome) of *P. lactiflora*. Similar to the chloroplast genome (cpgenome), the mitogenome of *P. lactiflora* extends across 181,688 base pairs (bp). Its unique quadripartite structure results from a pair of extensive inverted repeats, each measuring 25,680 bp in length. The annotated mitogenome includes 27 protein-coding genes, 37 tRNAs, 8 rRNAs, and two pseudogenes (*rpl5*, *rpl16*). Phylogenetic analysis was performed to identify phylogenetic trees consistent with Paeonia species phylogeny in the APG Ⅳ system. Moreover, a total of 12 MTPT events were identified and 32 RNA editing sites were detected during mitogenome analysis of *P. lactiflora*. Our research successfully compiled and annotated the mitogenome of *P. lactiflora*. The study provides valuable insights regarding the taxonomic classification and molecular evolution within the Paeoniaceae family.

## 1. Introduction

*Paeonia lactiflora* Pall. (*P. lactiflora)*, belonging to the genus *Paeonia* L. (Garl von Linne, 1935) of the *Paeoniaceae* Raf. family, is an iconic floral emblem of the Tanabata festival. The germplasm of *Paeonia* species holds significant value to both the floral and pharmaceutical industries [1]. Yangzhou, Zhongjiang, Bozhou and Heze have emerged as the four leading producers of *P. lactiflora* in China, with the plant being cultivated in nearly every corner of the aforementioned cities [2]. Moreover, these plants boast substantial medicinal potential. Paeoniaceae species are rich in compounds like monoterpene glycosides, flavonoids, tannins, stilbenes, and triterpenoids, among others [3]. According to traditional Chinese medicine, *P. lactiflora* aligns with yin, and is considered effective for nourishing yin [4], alleviating heat, and mitigating pain [5]. As a perennial plant, its dried roots serve as a popular ingredient in traditional remedies.

Mitochondria, widely recognized as the powerhouses of the cell, are critical for eukaryotic life [6]. While they have their own genome, most mitochondrial proteins are imported from the nuclear genome. Contrary to animal mitogenomes, plant mitogenomes demonstrate significant differences in size and genomic organization [7]. Mitogenomes typically take the form of naked, double-stranded DNA molecules, occurring predominantly as loops but also as linear molecules, with sizes varying across species [8]. Repetitive sequences, which make up the majority of the genome in many species [9], and their complex structures complicate the sequencing and assembly of mitogenomes compared to plastid genomes. The transmission patterns of mitogenomes are distinct from those of nuclear genomes [10]. The lack of histone protection and a DNA damage repair system make the mitogenome more susceptible to mutations [11]. Additionally, assembling repetitive regions and obtaining complete mitogenome sequences are challenging tasks with short-read sequencing data from platforms like Illumina [12,13,14]. With the advent of long-read sequencing technologies by PacBio and Oxford Nanopore in recent years, sequencing plant mitogenomes has become more feasible and cost-effective [15,16].

A few years ago, the pioneering sequencing of the genus *Paeonia*’s genome was undertaken, with several genomic resources now available for species including *Paeonia jishanensis* T. Hong and W. Z. Zhao (*P. jishanensis)*, *Paeonia delavayi* Franch. (*P. delavayi)*, *Paeonia sinjiangensis* K. Y. Pan (*P. sinjiangensis)*, and several native variants [17]. This endeavor significantly enriched the genetic resources for *Paeonia* varieties, offering a wealth of data to breeders. Despite these advancements, systematic studies on the mitogenome of *P. lactiflora* have been scarce. Only one mitogenome resource of a *Paeonia* species has been published, with its details yet to be extensively examined [18], thereby limiting our understanding of mitogenome evolution in the Paeoniaceae family. Homologous recombination, a common occurrence in plant mitogenomes, often leads to frequent duplication and replacement of spacer regions [19]. This process enables the merging of certain genetic information from different species, paving the way for innovative methodologies to study the origins and evolution of angiosperms, DNA repair, the maintenance of genetic diversity, and the emergence of heterosis [20].

RNA editing, particularly in the form of C to U base conversions, is a frequent phenomenon in organelles like mitochondria and chloroplasts [21,22,23]. This process in plant mitochondria necessitates the involvement of a multitude of nuclear-encoded factors [21]. Many C→U (and U→C in some plants) conversions modify the coding sequence of numerous organellar transcripts [24], facilitating translatable mRNA production through the generation of an AUG start site or elimination of premature stop codons, or altering RNA structure, splicing, and stability. By generating new start or stop codons or modifying the amino acid sequence, RNA editing fosters the formation of the correct secondary structure of proteins, enabling their functionality and ensuring the precise expression of genetic information within cells [25,26,27].

Organelle DNA transfer into the nucleus occurs in the form of ongoing transfer of mitogenome or cpgenome fragments into the nuclear genome. Transferred organelle genome mainly forms non-coding sequences or pseudogenes in the nuclear genome, and these fragments are called nuclear mitochondrial DNA (NUMT) [28] and nuclear chloroplast DNA (NUPT) [29], respectively. At present, NUMT and NUPT have been found in the nuclear genomes of most eukaryotes, and a large part of them are associated with nuclear genes [30]. DNA transfer between organelles is another type of intracellular gene transfer. Mitochondrial plastid DNAs (MTPTs) refers to the chloroplast-derived DNA that is contained in the mitogenome of angiosperms, representing 0.1% to 10.3% of the mitogenome [31]. The analysis of MTPTs will lead to complex mitogenome results, as most mtpts evolution is relatively unconstrained by function; the MTPT’s sequence evolution reveals the possibility of mutations in the mitogenome of angiosperms [32].

In this study, we sequenced the mitogenome of a *P*. *lactiflora* cultivar to elucidate its genome structure, analyzed repeat sequences and investigated genomic recombination events. Furthermore, we aimed to conduct a comparative analysis with other mitogenomes within the Paeoniaceae family. These findings enhance our understanding of organelle genome evolution in *P. lactiflora* and lay a solid foundation for future breeding research, helping to unravel the phylogenetic relationships of species based on organelle genomes and the transference of DNA between organelle genomes [33]. Hybridization among diverse *P. lactiflora* species can overcome barriers, thereby augmenting genetic diversity [34].

## 2. Materials and Methods

### 2.1. Plant Materials, DNA and RNA Extraction, and Sequencing

The fresh juvenile leaves were provided using plants housed in the Botanical Garden of Medicinal Plants in Beijing, China (E116°25′, N39°47′). The samples were rinsed and cleaned with DEPC water, then preserved at −80 °C until further use. Total genomic DNA was extracted using the Cetytrimethylammonium Bromide (CTAB) method [35], followed by the utilization of a plant genomic DNA kit (Tiangen Biotech, Beijing, Co., Ltd., Beijing, China). The same DNA sample was employed for both Illumina sequencing and Oxford Nanopore sequencing. As part of the study, 350 bp fragments were incorporated into a DNA library and sequenced on the Illumina HiSeq X platform (Benagen Technology Co., Ltd., Wuhan, China) for short-read sequencing. For Oxford Nanopore sequencing, the DNA library was constructed using the DNA Library Kit with the catalog number SQK-LSK110.

### 2.2. Mitochondrial Genome Assembly and Annotation

To extract the complete sequence and investigate the potential structure of the mitogenome of *P. lactiflora*, a hybrid assembly strategy was employed. Short mitogenome reads were extended using Getorganelle (v1.6.4) [36], and the resulting readings were assembled into a unitig graph using SPAdes software included in Unicycler (v0.4.9). The repeat region in the unitig graph was resolved using Unicycler with Nanopore long reads. The assembled mitogenome of *P. lactiflora* was annotated using the GeSeq web server [37] and IPMGA. Any issues with the protein-coding genes (PCGs) were checked and rectified using Apollo, and the genome map was constructed with OGDRAW [38]. All tRNA genes were verified using the default parameters of tRNAscan-SE [39].

### 2.3. Repeat Sequence Analysis and MTPT Prediction

The parameters for identifying repeat sequences were specified by using MISA [40] with “1-10 2-5 3-4 4-3 5-3 6-3”. Additionally, Tandem Repeats Finder (TRF) [41], and ROUS Finder were also used [42]. TRF was employed to identify tandem repeats with the parameters “2 7 7 80 10 50 500 –f -d –m”, while ROUS Finder, with the default parameters “-o input_repeat m 24 -b /usr/bin/ -k false -gb false”, was used to identify dispersed repeats.

For an accurate analysis of mitochondrial plastid DNA (MTPTs), the cpgenome was assembled using Illumina reads. The GetOrganelle software [23], with default parameters “get_organelle_from_reads.py -1 forward.fq -2 reverse.fq -o plastome_output -R 15 -k 21,45,65,85,105 -F embplant_pt”, was used to assemble the cpgenome from the same mitogenome assembly data. The cpgenome was annotated using CPGAVAS2 [43], visualized using CPGView [44], and any annotation errors flagged via CPGView were manually corrected with Apollo. MTPTs were identified using BLASTN based on the assembled cpgenome that is longer than 100 bp.

### 2.4. Phylogenetic Analysis

Given the large non-conservation of non-coding regions in plant mitogenomes, we constructed trees using common genes. First, 28 mitogenomes were downloaded from the NCBI database, and common genes (*atp6*, *atp8*, *ccmC*, *ccmFC-e2*, *ccmFN*, *cox3*, *matR-e1*, *matR-e2*, *nad1-e1*, *nad1-e5*, *nad3*, *nad4-e1*, *nad4-e2*, *nad4-e3*, *nad4-e4*, *nad5-e2*, *nad6*, *nad7-e1*, *nad7-e3*, *nad7-e4*, *nad7-e5*, *nad9, adh4*) were extracted using PhyloSuite [45]. These common protein-coding regions underwent multiple sequence alignment using MAFFT v7.505 software [46]. The resulting sequences were linked in tandem with PhyloSuite and analyzed using IQTREE [47] for phylogenetic purposes. The final phylogenetic analysis results were visualized using ITOL v6.8.1 software [48].

### 2.5. RNA Editing Site Identification and Validation

To identify RNA editing sites, we extracted both DNA and RNA from the whole genome. We used PhyloSuite [45] software to extract the protein-coding genes (PCGs) of the studied taxa. For our prediction of C-to-U RNA editing events in plant mitochondria, we employed Deepred-Mt [49], a program that uses deep characterization learning. This program used default parameters to predict each PCG for the selected species. 

## 3. Results

### 3.1. Elucidating Mitogenome Structure and Broad Genomic Features through Graph-Based Techniques

In order to assemble the mitogenome of *P. lactiflora*, we implemented a hybrid assembly strategy that capitalized on both long and short-read sequencing technologies. From the total DNA, we sequenced and assembled the mitogenome length of 181,688 bp. Leveraging the GetOrganelle v1.7.7.0 software, we isolated and expanded the mitogenome short reads. The genome was assembled into a graph-based mitogenome (Figure 1A); these reads were then assembled into a unitig graph with the help of the SPAdes software embedded within the Unicycler software suite, thus establishing the graph model of *P. lactiflora*’s mitogenome (Figure 1D).

The constructed graph consists of fifteen contigs, three of which exhibit a double bifurcation (Figure 1). Contig 5 stands out as the longest, stretching to 51,364 bp, while repeat2-1 and repeat2-2 remain the shortest, amounting to only 181 bp (Table 1). Importantly, the complex double bifurcating structures were resolved using the Nanopore long-reads, and all contigs were subsequently merged with the Bandage software. In conclusion, the mitogenome of *P. lactiflora* is encapsulated within a single chromosome spanning 181,688 bp, with a GC content of 55.7%. The GC content of its related species was calculated; the GC content of *Heuchera parviflora* var. *Saurensis* R.A.Folk and *Sedum plumbizincicola* X.H.Guo and S.B.Zhou ex L.H.Wu were 47.18% and 45.93%, respectively, while the GC content of *Rhodiola crenulata* (Hook.f. and Thomson) H.Ohba was 46.64%, *Ribes nigrum* L. was 47.25%, *Loropetalum chinense* (R.Br.) Oliv. was 47.35%, *Rhodiola tangutica* (Maxim.) S.H.Fu was 45.67%, and the GC content of *Rhodiola juparensis* (Fröd.) S.H.Fu was 46.01%.

### 3.2. Genome Annotation

We conducted a comprehensive annotation of the mitogenome of *P. lactiflora*, which includes 27 protein-coding genes (PCGs), 17 transfer RNA genes (tRNA), and 3 ribosomal RNA genes (rRNA).

The classification of these PCGs based on their function is elaborated in Table 2. The mitogenome of *P. lactiflora* was confirmed to possess annotations for all core genes, including five small subunit ribosomal genes (*rps3*, *rps4*, *rps10*, *rps12*, *rps14*), two large subunit ribosomal genes (*rpl5*, *rpl16*), and a solitary succinate dehydrogenase subunit gene (*sdh4*) as shown in Figure 2.

### 3.3. Repeat Element Analysis

Simple Sequence Repeats (SSRs), also known as tandem repeats, are prevalent components in all known genomes [50]. They are considered among the most significant genetic markers and can be found in the majority of species [51]. In this investigation, we used the MISA software to predict SSRs in the mitogenome of *P. lactiflora*. As a result, we identified a total of 55 SSRs on the mitogenome’s chromosome, which included 15 monomers and dimers, 12 trimers, 8 tetramers, 17 pentamers, and 3 hexamers (Appendix A). These SSR loci have the potential to serve as valuable molecular markers in future research.

Tandem Repetitive Sequences (TRS) are contiguous, repeating units with a high degree of sequence identity [52]. They have been implicated in numerous genetic processes, such as chromosomal organization, gene regulation, and disease diagnosis [53]. Using Tandem Repeats Finder software, we predicted tandem repetitive sequences in the mitogenome of *P. lactiflora*. As shown in Appendix A, we identified a total of 26 TRS, with TRS18 possessing the longest repeat unit at 45 bp and TRS4, TRS18, and TRS22 exhibiting the greatest cumulative lengths at 98 bp. 

Dispersed Repetitive Sequences are DNA sequences that are repeated throughout the genome and play crucial roles in genome rearrangement, evolution, and stress resistance [10]. We used ROUSFinder software to predict dispersed repetitive sequences in the mitogenome of *P. lactiflora*, identifying two types of matching directions (Appendix A). The chromosome with the longest dispersed repetitive sequence, named DRS1, measured 224 bp in length. However, further analysis and experimental evidence are required to determine whether these repetitive sequences can mediate homologous recombination (Figure 3).

### 3.4. Mitochondrial Plastid DNAs Prediction

MTPTs are remnants of plastid-derived DNA in mitogenomes [54], which means that they are fragments of DNA that originated from the cpgenome and have been incorporated into the mitogenome. In this study, the researchers assembled the cpgenome of *P. lactiflora* (PP049084) using the same dataset as the mitogenome assembly. The cpgenome spanned 152,731 bp and had a GC content of 38.00%; it contained 85 genes, 112 of which were unique, including 87 protein-coding genes (79 unique), 37 tRNA genes (29 unique), and 8 rRNA genes (4 unique). According to the cpgenome, we identified nine MTPTs between the mitogenome (NC_070189.1) and cpgenome of *P. lactiflora* via sequence similarity calculations, as delineated in Appendix A. These MTPTs constituted 2.2% of the total mitogenome. MTPT1, MTPT2 (1683 bp) was the longest, while MTPT10 (77 bp) was the shortest identified MTPT. The genomic positions of these MTPTs in the cpgenomes and mitogenomes are presented in Table 3 and Figure 3. To delve into their functions, we annotated the DNA fragments of the 12 MTPTs, identifying 10 unique genes in total. Two protein genes and six tRNA genes (*rps19, petG, trnW-CCA, trnP-UGG, trnD-GUC, trnH-GUG, trnM-CAU, and trnI-CAU*) were fully intact, while three genes (*rpl2, rpl16, and rrn16s*) became pseudogenes during migration. Among the MTPTs identified in the mitogenome of *P. lactiflora*, MTPT1, MTPT2, MTPT4, MTPT5, and MTPT11 are located in repeat sequences, including genes such as rpl23 and rpl2. These genes are hotspots for gene transfer [55,56,57].

To explore the MTPTs pattern in *P. lactiflora* and its homologous species, we searched the NCBI database and found that the mitogenome data of seven closely-related species within the Saxifragales Order have been published, including *Loropetalum chinense* var. rubrum Yieh, *Heuchera parviflora* var. saurensis R.A.Folk, *Sedum plumbizincicola*, *Rhodiola crenulata* (Hook. f. et Thoms.) H. Ohba, *Ribes nigrum* L., *Rhodiola tangutica* (Maximowicz) S. H. Fu, and *Rhodiola coccinea* (Royle) Borissova (Appendix A). In the results of the MTPTs analysis of these seven closely-related species, fragments were found in repeat sequences, including the species with the most gene numbers, *H. parviflora* [58,59]. The number of MTPTs in Saxifragales, which belongs to the same order as *P. lactiflora*, is the least with only 11, particularly compared to the most in *L. chinense*, which has a significant difference. This may suggest that external factors such as the environment, human intervention, and metabolic influences have affected the evolutionary process of closely-related species [60].

### 3.5. Phylogenetic Relationships

We conducted phylogenetic analyses by aligning 25 common genes from 29 angiosperm species, including a broad spectrum of species such as *Vitis vinifera* L. (NC_012119.1), *Zygophyllum fabago* L. (MK431827.1), *Tribulus terrestris* Muhl. (MK431825.1), *Eucalyptus grandis* W.Hill ex Maiden (NC_040010.1), *Oenothera biennis strain suaveolens* Grado (MZ934756.1), *Lagerstroemia indica* L. (NC_035616.1), *Oenothera elata subsp. hookeri strain johansen Standard* (MZ934757.1), *Oenothera villaricae strain berteriana Schwemmle* (MZ934755.1), *Medinilla magnifica* Lindl. (MT043351.1), *Geranium maderense voucher* (NC_027000.1), *Acacia ligulata* A.Cunn. ex Benth. (NC_040998.1), *Medicago truncatula* Gaertn. (NC_029641.1), *Gleditsia sinensis* Lam. (NC_058235.1), *Glycine soja* Siebold and Zucc. (NC_039768.1), *Glycyrrhiza uralensis* Fisch. ex DC. (NC_053919.1), *Haematoxylum brasiletto* H.Karst. (NC_045040.1), *Ormosia boluoensis* Y.Q.Wang and P.Y.Chen (NC_059804.1), *Phaseolus vulgaris* L. (NC_045135.1), *Pisum abyssinicum* A.Braun (NC_059791.1), *Senna tora* (L.) Roxb. (NC_038053.1), *Glycine max* (L.) Merr. (NC_020455.1), *Vigna radiata* (L.) R.Wilczek (NC_015121.1), *Sophora flavescens* Aiton (NC_043897.1), *Stylosanthes pilosa* M.B.Ferreira and Sousa Costa (NC_063516.1), *Tamarindus indica* L. (NC_045038.1), *Trifolium grandiflorum* Hook. and Arn. (NC_048501.1), *β macrocarpa* Guss. (NC_015994.1), *Mirabilis jalapa* L. (NC_056991.1), and, of course, *P. lactiflora*. By utilizing these common PCGs within the mitogenomes, we generated a phylogenetic tree using IQTREE2 (v1.2.3). Figure 4 clearly demonstrates a distinct clade formed by the relationship between *P. lactiflora* and *V. vinifera*. The complete mitogenome of *P. lactiflora* serves as a valuable asset in developing molecular markers and enhancing our understanding of the evolutionary history and cultivation strategies of Paeoniae.

### 3.6. Prediction of RNA-Editing Sites

By mapping transcriptome data onto the mitogenomes, we were able to identify 32 sites of RNA modification within the spacer and protein-coding regions [61]. These modifications were found within a diverse range of genes, including *atp1*, *atp4*, *atp6*, *atp8*, *atp9, ccmB*, *ccmC*, *ccmFC*, *ccmFN*, *cob*, *cox1*, *cox2*, *cox3*, *matR*, *mttB*, *nad1*, *nad2*, *nad3*, *nad4*, *nad4L, nad5*, *nad6*, *nad7*, *nad9*, *rpl16*, *rpl5*, *rps12*, *rps4*, and *sdh4* (Figure 5). A total of 569 RNA editing events occurred, with the most editing events occurring in nad4, with 47 edits, and the fewest in *rps7* and *rps14*, with 2 edits each. The proportion of editing events with more than 20 edits was 34.4%. Out of the identified modifications, there were 747 instances with a frequency greater than 0.01, with 289 of these boasting a certainty of 1 [62]. The proportion of editing events with a frequency greater than 0.5 is 99.07%, indicating that there are many gene-editing events. There was a total of 13 types of amino acid change events, with the most frequent being the conversion of proline to leucine, which occurred 147 times. The majority of RNA editing events that did not result in changes in amino acid type were synonymous mutations, where the base substitution did not alter the amino acid. There were three events with more than 100 times, with the other two types being the conversion of serine to phenylalanine and phenylalanine to leucine [63].

## 4. Discussion

The recent publication of our study on the mitogenome of *P. lactiflora*, utilizing both Nanopore long reads [62] and Illumina short reads [64], marked the first-ever confirmation that the mitogenome of this species primarily comprises circular chromosomes. To gain a comprehensive understanding of the diversity and evolution of *P. lactiflora*’s mitogenome, we conducted extensive sequencing, assembly, and detailed characterization. This involved the determining the gene content, simple sequence repeats (SSRs), tandem repeats, dispersed repeats, mitochondrial plastid DNAs (MTPTs), and RNA-editing events within *P. lactiflora*’s mitogenome. The insights gained from this study are crucial for further exploration of the diversity and evolutionary dynamics of the *P. lactiflora* mitogenome. 

In this study, we sequenced and assembled the mitogenome of *P. lactiflora*, which has a circular genome structure of 181,688 bp. The mitogenome of another species in the Paeoniaceae family, *Paeonia suffruticosa* Andrews, has been published with a size of 203,077 bp (https://www.ncbi.nlm.nih.gov/nuccore/OR551751.1 (accessed on 13 February 2024)). These two species are currently the only ones in the Paeoniaceae family to have their mitogenomes sequenced, and their sizes are relatively small compared to the range of 66 kb to 11 Mb for mitogenomes in eudicots. According to the data on NCBI, the whole genome size of *Paeonia suffruticosa* Andrews is 12.28 Gb [65], while the mitogenome is relatively small, indicating that the gene fragment interaction between the mitogenome and the nuclear genome is relatively small.

Mitochondrial plastid DNA (MTPTs) refers to DNA fragments that have migrated from the plastid genome to the mitochondrial genome, belonging to a type of horizontal gene transfer (HGT). In this study, 11 MTPTs were discovered in *P. lactiflora* through sequence similarity between the cpgenome and mitogenome, with a total insertion length accounting for 3.6% of the mitogenome. Among the seven closely-related species analyzed, the cpgenome found in Heuchera parviflora var. saurensis was the most abundant, accounting for 8.5%, while the cpgenome found in Loropetalum chinense was the least abundant, accounting for 1.4%.

In this study, we discovered 569 potential RNA editing sites in the mitogenome of *P. lactiflora*, leading to 13 distinct types of amino acid substitutions. The RNA editing events were least prevalent in ribosomal protein genes like *rps4* and *rps14*, whereas genes such as *nad4*, *ccmB*, and *ccmFn* exhibited a higher frequency of RNA editing events. These findings suggest that RNA editing plays a crucial role in plant adaptation to environmental changes and signal transduction, particularly in mitochondrial and plastid biogenesis [66]. 

Repetitive sequences are widely distributed in plant mitogenomes and have relatively complex structures. These sequences can cause changes in the structure of plant genomes through recombination, thereby influencing plant evolution. In the mitogenome of *P. lactiflora*, various repetitive sequences were observed, including 55 pairs of SSRs, 26 pairs of Tandem repeat sequences, 623 pairs of Palindromic repeats, and 641 pairs of Forward repeats. The recombination of these sequences has a significant impact on the size, gene arrangement, and evolution of plant mitogenomes and may also lead to phenotypic mutations in plants. For instance, in *Pisum sativum* L., the amplification of repetitive sequences results in seed mutations that affect subsequent growth and development [67]. In *Zea mays* L., repetitive sequences influence the expression of chloroplast genes, thereby affecting photosynthesis and growth [68]. In *Oryza sativa* L., variations in repetitive sequences can impact traits such as plant height and tiller number [69]. Furthermore, repetitive sequences play an important role in cytoplasmic male sterility. Some plant mitogenomes contain numerous repetitive sequences that can cause gene amplification, deletion, or rearrangement, thereby affecting pollen development. For example, in certain Saccharum officinarum varieties, the amplification of a mitogenome repeat sequence is associated with male sterility [70]. The analysis of the mitogenome of *P. lactiflora* provides a theoretical basis for understanding subsequent seedling breeding and phenotypic changes in *P. lactiflora*.

## Figures and Tables

**Figure 1 genes-15-00239-f001:**
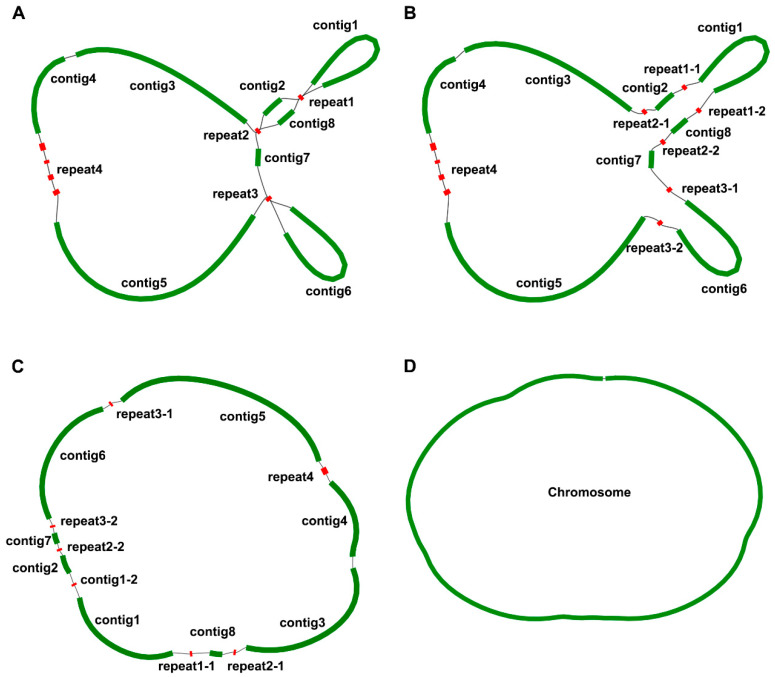
Schematic representation of the assembly results of *P. lactiflora* mitogenome. The green contigs represent a single copy of the mitogenome fragment, and the red contigs represent the repeat sequences. (**A**). The second generation data assembly of the De Bruijn Graph; (**B)**. Third Generation Data processing repeat sequences; (**C**). Adjust the position of each fragment sequence; (**D**). Graphical representation of the mitogenome of *P. lactiflora*.

**Figure 2 genes-15-00239-f002:**
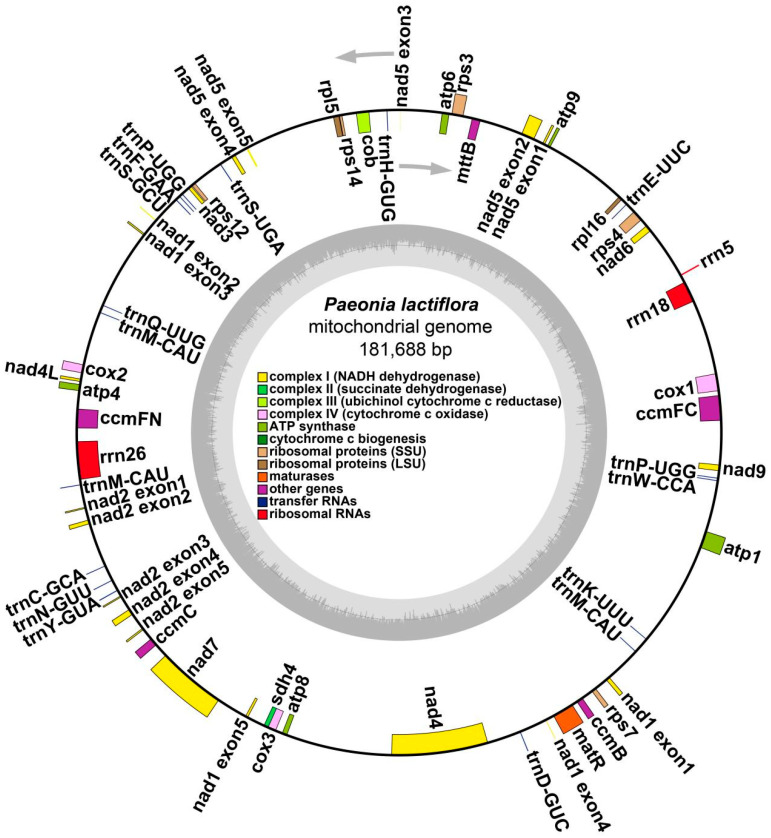
Schematic of the mitogenome of the *P. lactiflora*. Genes inside and outside the circle are transcribed clockwise and counterclockwise, respectively. Genes were shown in different colors based on their functional classification, which is shown in the center of the map.

**Figure 3 genes-15-00239-f003:**
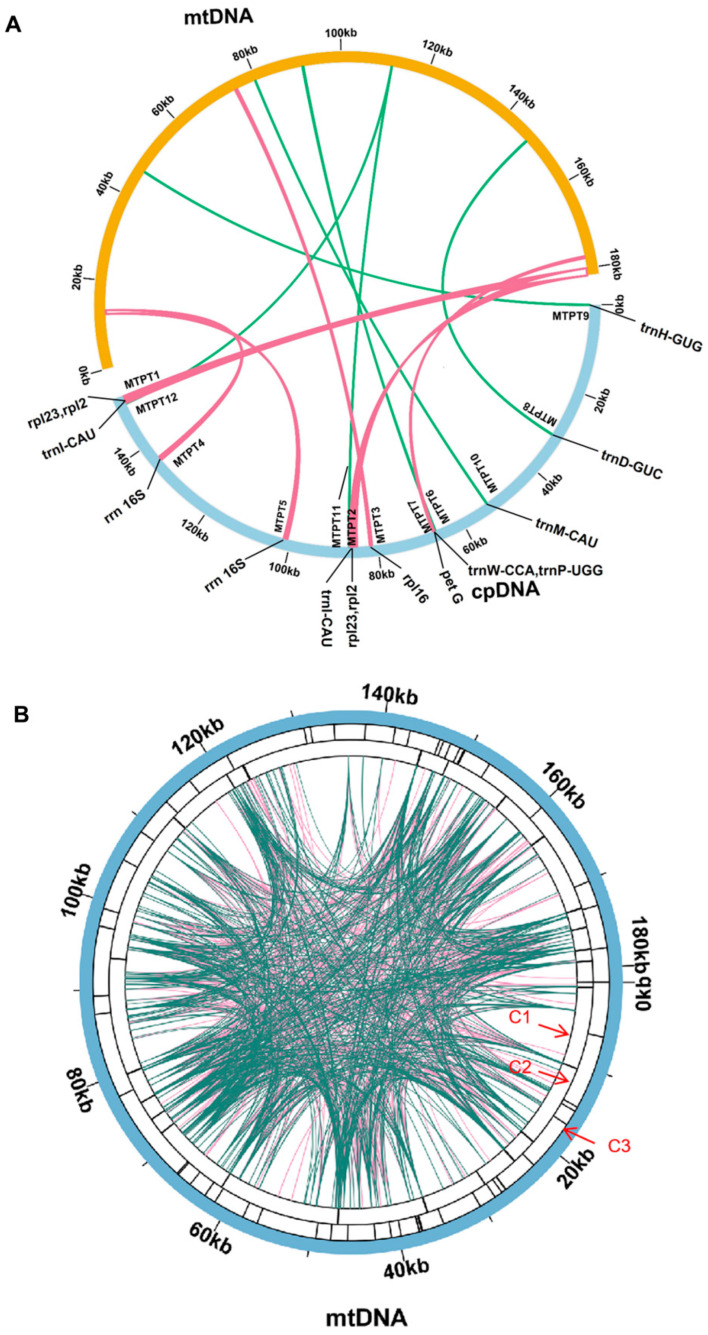
The repeat of the *P. lactiflora* organelle genome. (**A**) The yellow color represents the mitogenome, the blue color represents the cpgenome, and the inside connecting red and green color lines represent similar DNA. (**B**) The repeat sequences identified in the mitogenome. The C1 circle shows the dispersed repeats connected with blue arcs from the center from outward. The C2 circle shows the tandem repeats as short bars. The C3 circle shows the microsatellite sequences identified using MISA. The scale is shown on the C3 circle, with intervals of 20 kb.

**Figure 4 genes-15-00239-f004:**
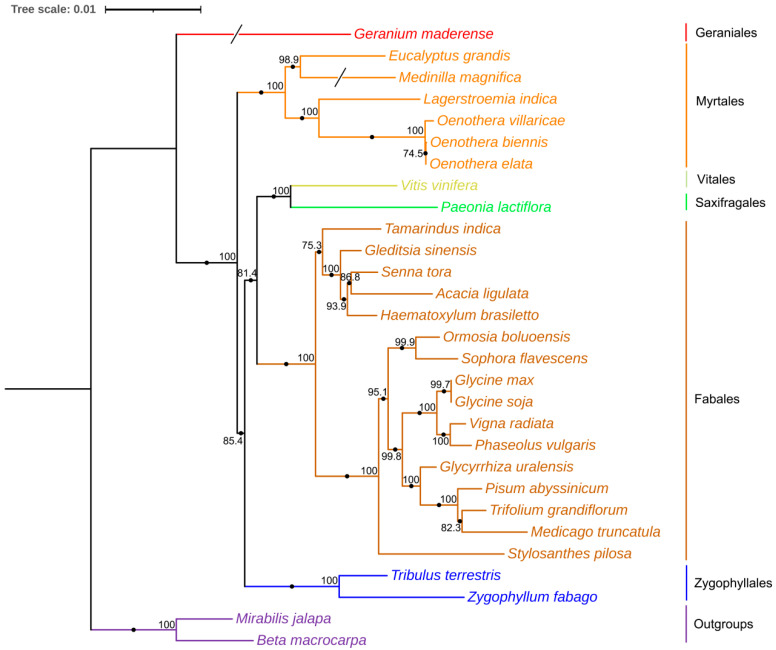
Phylogenetic tree based on 25 homologous protein-coding genes (*atp6*, *atp8*, *ccmC*, *ccmFC-e2*, *ccmFN*, *cox3*, *matR-e1*, *matR-e2*, *nad1-e1*, *nad1-e5*, *nad3*, *nad4-e1*, *nad4-e2*, *nad4-e3*, *nad4-e4*, *nad5-e2*, *nad6*, *nad7-e1*, *nad7-e3*, *nad7-e4*, *nad7-e5*, *nad9*, *adh4*) in 29 plants’ mitogenomes using maximum likelihood (ML) analysis. Numbers above nodes are support values with ML bootstrap values and branch lengths. The red color represents Geraniales, the orange color represents Myrtales, the yellow color represents Vitales, the green color represents Saxifragales, the brown color represents Fabales, the blue color represents Zygophyllales, the purple color represents Outgroups.

**Figure 5 genes-15-00239-f005:**
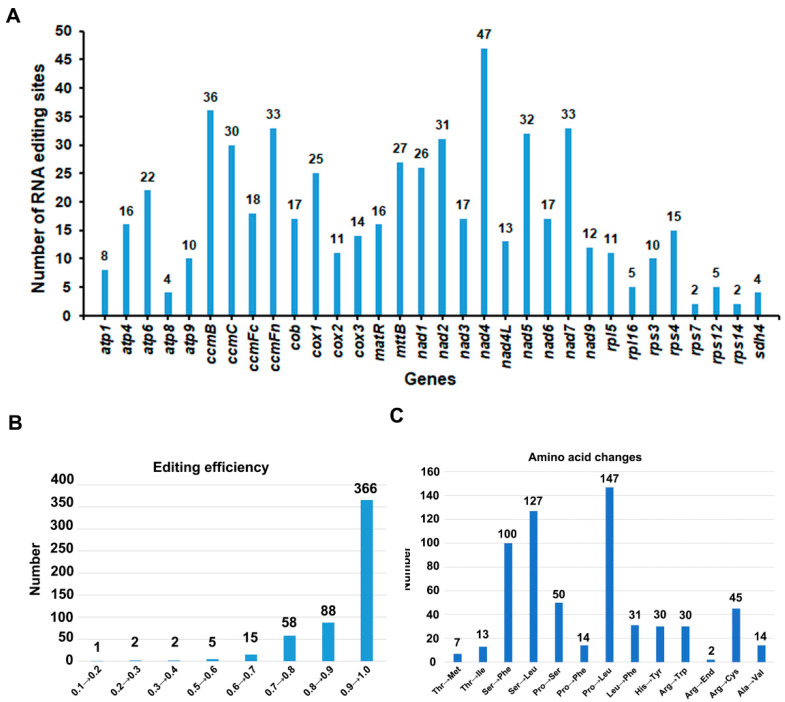
The characteristics of the RNA editing sites identified in protein-coding genes (PCGs). The “X” axis shows the name of protein-coding genes, and the “Y” axis shows the number of predicted RNA editing sites. (**A**) The number of RNA editing sites identified in each PCGs of mitogenome; (**B**) RNA editing efficiency; (**C**) Frequency of amino acid changes caused by RNA editing.

**Table 1 genes-15-00239-t001:** The length, bifurcation structures, and chromosome attribution of each assembled contig of *P. lactiflora* mitogenome.

No.	Name	Length (bp)	Chromosomes	Bifurcation Structures
1	contig1	31,787	Chromosome1	No
2	contig2	3211	Chromosome1	No
3	contig3	35,924	Chromosome1	No
4	contig4	16,986	Chromosome1	No
5	contig5	51,364	Chromosome1	No
6	contig6	31,950	Chromosome1	No
7	contig7	2735	Chromosome1	No
8	contig8	4860	Chromosome1	No
9	repeat1-1	224	Chromosome1	Yes
10	repeat1-2	224	Chromosome1	Yes
11	repeat2-1	181	Chromosome1	Yes
12	repeat2-2	181	Chromosome1	Yes
13	repeat3-1	189	Chromosome1	Yes
14	repeat3-2	189	Chromosome1	Yes
15	repeat4	1683	Chromosome1	No

**Table 2 genes-15-00239-t002:** Gene composition in the mitogenome of *P. lactiflora*.

Group of Genes	Name of Genes
ATP synthase	*atp1*, *atp4*, *atp6*, *atp8*, *atp9*
NADH dehydrogenase	*nad1*, *nad2*, *nad3*, *nad4*, *nad4L*, *nad5*, *nad6*, *nad7*, *nad9*
Cytochrome b	*cob*
Cytochrome oxidase	*cox1*, *cox2*, *cox3*
Maturases	*matR*
Protein transport subunit	*mttB*
Other genes	*ccmFC*, *ccmB*, *ccmC*, *ccmFN*
Ribosomal protein large subunit	*rpl5*, *rpl16*
Ribosomal protein small subunit	*rps3*, *rps4*, *rps7*, *rps12*, *rps14*
Succinate dehydrogenase	*sdh4*
Ribosomal RNA genes	*rrn5*, *rrn18*, *rrn26*
Transfer RNA genes	*trnC-GCA*, *trnD-GUC*, *trnE-UUC*, *trnF-GAA*, *trnM-CAU*, *trnH-GUG*, *trnK-UUU*, *trnN-GUU*, *trnP-UGG*(*×2*), *trnQ-UUG*, *trnS-GCU*, *trnW-CCA*, *trnY-GUA*, *trnS-UGA*, *trnM-CAU*(*×2*)

**Table 3 genes-15-00239-t003:** Plastid homologous sequences identified in mitogenome (MTPTs) of *P.lactiflora*.

Number	Alignment Length (bp)	Mitogenome		Cpgenome		MTPT Annotation
		Start	End	Start	End	
MTPT1	1683	181,688	180,006	150,737	152,397	Complete (*rpl23*), Partial (*rpl2*)
MTPT2	1683	180,006	181,688	84,737	86,397	Complete (*rpl23*), Partial (*rpl2*)
MTPT3	504	75,234	74,739	81,290	81,791	Partial (*rpl16*)
MTPT4	886	13,207	12,349	135,451	136,315	Partial (*rrn16S*)
MTPT5	886	12,349	13,207	100,819	101,683	Partial (*rrn16S*)
MTPT6	408	177,268	177,662	66,634	37,018	Complete (*trnW-CCA, trnP-UGG*)
MTPT7	212	91,080	91,288	66,378	66,583	Complete (*petG*)
MTPT8	142	147,487	147,346	31,196	31,332	Complete (*trnD-GUC*)
MTPT9	79	46,463	46,541	1	79	Complete (*trnH-GUG*)
MTPT10	77	79,847	79,771	53,073	53,149	Complete (*trnM-CAU*)
MTPT11	79	111,678	111,604	86,457	86,535	Complete (*trnI-CAU*)

## Data Availability

The assembled mitochondrial genome sequences have been deposited in NCBI (https://www.ncbi.nlm.nih.gov/ (accessed on 25 January 2023)) with accession number: NC_070189.1; The assembled chloroplast genome sequences have been deposited in NCBI with accession number: PP049084.1; The BioProject ID: PRJNA905540 (https://www.ncbi.nlm.nih.gov/bioproject/905540 (accessed on 10 February 2024)); The BioSample ID: SAMN37366985 (https://www.ncbi.nlm.nih.gov/biosample/37366985 (access on 10 February 2024)); SRA ID: SRR15412863 (https://trace.ncbi.nlm.nih.gov/Traces/index.html?view=run_browser&acc=SRR15412863&display=metadata (access on 10 February 2024)).

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
