# Peer review of "The Complete Mitochondrial Genome of Paeonia lactiflora Pall. (Saxifragales: Paeoniaceae): Evidence of Gene Transfer from Chloroplast to Mitochondrial Genome"

_genes, 2024, doi:10.3390/genes15020239_

Round 1

Reviewer 1 Report

Comments and Suggestions for Authors

My general considerations are:

When we mention the family, genus and species for the first time, we must include the authors;

Italicize the names of the species; The authors weren't very careful about this.

Check the legend in Figure 2, very repetitive. I suggested a new text.

Author Response

Dear professor! Please see the attachment.

Reviewer 2 Report

Comments and Suggestions for Authors

The author presents a comprehensive analysis of the mitochondrial genome (mitogenome) of Paeonia lactiflora, with a particular focus on gene transfer from the chloroplast to the mitochondrial genome. While the current version is detailed and methodologically sound, I do want to raise a few points:

1) Avoid repetitions: "Moreover, 32 RNA-editing sites..." "....identified and 32 RNA editing sites were..."

2) Given the hybrid assembly strategy, how effectively can the authors guarantee that any discrepancies between them are properly resolved?

3) "and is considered effective for nourishing Yin [4]" I didn't find anything in the citation stating what was cited.

4) "We used PhyloSuite (REF)" several references are missing in this section.

5) "All data generated by this study are available at the corresponding author upon
reasonable request." What this means? All the data should be deposited in a repository; it shouldn't be needed as a request; and even more strange, what is the classification of a'reasonable request'? Science should be open and the reproducible, all the data should be shared.

Author Response

(The authors gave the same response as above.)
